# Comparative Transcriptomic Analysis Between High- and Low-Growth-Rate Meat-Type Rabbits Reveals Key Pathways Associated with Muscle Development

**DOI:** 10.3390/ani15111585

**Published:** 2025-05-29

**Authors:** Chao Yang, Lingxi Zhu, Li Tang, Xiangyu Zhang, Min Lei, Xiaohong Xie, Cuixia Zhang, Dingsheng Yuan, Congyan Li, Ming Zhang

**Affiliations:** 1Key Laboratory of Animal Genetics and Breeding in Sichuan Province, Sichuan Academy of Animal Science, Chengdu 611000, China; yc20040736@163.com (C.Y.); tanglijiean@263.com (L.T.); zhangxiangyu75@163.com (X.Z.); meilei@vip.126.com (M.L.); xiexiaohong@vip.163.com (X.X.); zhangcuixia@163.com (C.Z.); yuandingsheng@163.com (D.Y.); 2College of Animal Science and Technology, Sichuan Agricultural University, Chengdu 611130, China; zhulingxi1219@163.com; 3State Key Laboratory of Swine and Poultry Breeding Industry, College of Animal Science and Technology, Sichuan Agricultural University, Chengdu 611130, China; 4Farm Animal Genetic Resources Exploration and Innovation Key Laboratory of Sichuan Province, Sichuan Agricultural University, Chengdu 611130, China

**Keywords:** muscle fiber, muscle development, transcriptome, rabbit

## Abstract

Rabbit meat represents a nutrient-dense, protein-rich food source experiencing increasing demand in the Asia-Pacific region. To investigate muscle growth differences between Sichuan linen (Ma) and Checkered Giant (Ju) rabbits, this study examined muscle fiber development in thigh and longissimus dorsi muscles at 28, 56, and 84 days post-natal. Significant differences in the muscle fiber area were observed between Ma and Ju rabbit at 56 days. Gene expression profiles of 56-day-old muscles were identified, followed by analysis of differentially expressed genes (DEGs) associated with muscle growth development. Functional analyses revealed critical biological pathways including glycolysis and HIF-1 signaling, as well as cellular processes such as muscle cell migration and programmed cell death. As the first comparative investigation of muscle development and transcriptional profiles between high- and low-growth-rate meat-type rabbits, this study elucidates molecular mechanisms underlying muscle growth, while offering a foundation for genetic enhancements aimed at optimizing meat production traits.

## 1. Introduction

As a lean and nutrient-dense protein source, rabbit (Oryctolagus cuniculus) meat has gained considerable attention in the agricultural animal production system. Global rabbit meat yield reached approximately 900,000 metric tons in 2021 [1]. The hindlimb serves as the primary depot for skeletal muscle growth in rabbits [2], and it also constitutes a critical determinant of meat nutritional quality [3]. Concurrently, the *longissimus dorsi* muscle, a key indicator of both meat quality parameters and growth kinetics, has been extensively investigated due to its significance in meat production economics and myogenesis research [4]. Consequently, these two muscles (thigh and *longissimus dorsi*) are frequently employed as model systems for studies examining muscle development mechanisms, meat quality attributes, and fiber type characteristics.

RNA sequencing (RNA-Seq) has emerged as a predominant methodology for transcriptomic investigations [5,6,7,8]. The continuous refinement of sequencing platforms has concurrently driven technological advancement and cost efficiency, and RNA-Seq has been used for the simultaneous quantification of differentially expressed genes (DEGs) across different experimental conditions to reveal molecular mechanisms [5,9,10]. Consequently, the high-throughput approach has been increasingly implemented to elucidate pivotal regulatory genes governing fundamental biological processes including developmental biology, metabolic regulation, and pathophysiological mechanisms [11,12,13,14,15].

The Sichuan linen (Ma) rabbit is an indigenous genetic resource in Sichuan Province, with advantageous traits including precocious sexual maturity, robust adaptability to coarse feeding regimens, and excellent meat quality. In contrast, the Checkered Giant (Ju) rabbit, a French-originated domestic breed recognized as one of the largest rabbit varieties, exhibits substantial body mass and high-yield characteristics in meat production [16]. Current scientific evidence indicates that differential gene expression patterns underlie phenotypic variations in skeletal muscle characteristics including fiber type composition, oxidative capacity, and insulin receptor binding. Particularly, skeletal muscle functionality is modulated by the expression dynamics of myosin heavy chain (MyHC) isoforms (I, IIa, IIx, IIb) [17], actin (ACTN) [18] and troponin C (TNNC) [19], which collectively determine myofiber structural and functional properties. A complex regulatory network encompassing myostatin (MSTN) [20], muscle regulation factors (MRFs) [21], paired box (PAX), myogenic transcription factor 2C (MEF2C), and insulin-like growth factors (IGFs) [22] coordinately governs skeletal muscle growth, myofiber type conversion, and intramuscular lipid metabolism [23]. Emerging transcriptomic analyses across animal models have partially characterized the transcriptional circuitry regulating fiber type, myofiber typing, hyperplasia, and hypertrophy [24]. The objective of the study is to comparatively analyze thigh and *longissimus dorsi* muscle transcriptomes in Ma and Ju rabbit, thereby identifying the hub-gene and primary signaling pathways responsible for their divergent muscle development patterns.

## 2. Materials and Methods

### 2.1. Animals and Muscle Tissue Samples

Experimental rabbits were individually housed in standardized cages with ad libitum water access and manual feeding protocols. Ma (*n* = 9) and Ju (*n* = 9) rabbits were fed with complete pelleted diets (Table 1) administered at 3% body weight daily to meet to dry matter intake requirement. Muscle specimens were obtained through standardized anatomical sampling; 0.5–1 cm^3^ of thigh muscle tissues was collected from the midpoint of the left quadriceps femoris, while 0.5–1 cm^3^ of *longissimus dorsi* muscle tissues was collected from the lumbar segments L3–L5 (1.5 cm right lateral to the spinal column) [25]. Tissue collection occurred at three developmental time-points (28, 56, and 84 days old) from both breeds. Specimens underwent triple PBS rinsing followed by bifurcated processing; one aliquot was flash-frozen in liquid nitrogen for subsequent RNA isolation, while the counterpart was fixed in 10% neutral-buffered formalin for histopathological preparation. This experimental protocol received formal approval from the Animal Ethical and Welfare Committee (AEWC) of Sichuan Agricultural University (Protocol No. DKY2023302115).

### 2.2. Paraffin Section Preparation and Statistical Analysis

Muscle tissue samples were fixed in 10% neutral-buffered formalin at room temperature for 72 h, subsequently dehydrated, embedded with paraffin, and sectioned at 4 μm thickness for hematoxylin–eosin (H&E) staining according to standard protocols [26]. Histological images were captured using an Evident APX100 imaging system (Olympus, Tokyo, Japan). Muscle fiber cross-sectional area (CSA) quantification was conducted by analyzing three non-overlapping visual fields per sample using image J software (Java 8) (NIH, Bethesda, MD, USA). Statistical comparisons of mean CSA values across developmental stages and breeds were performed through one-way ANOVA implementation in IBM SPSS Statistics version 27.0.1.

### 2.3. RNA Sample Detection and Library Construction

Total RNA was isolated from muscle tissue using TRNzol Universal Reagent (DP424, Tiangen Biotech, Beijing, China). RNA integrity was verified through electrophoretic analysis using the Bioanalyzer 2100 system (Agilent Technologies, Santa Clara, CA, USA). Polyadenylated mRNA was enriched from total RNA through poly-T oligonucleotide-conjugated magnetic bead selection. cDNA library preparation was conducted through the following sequential procedures: (1) fragmentation of mRNA templates using divalent cations in First Strand Synthesis Reaction Buffer (5×) under thermal incubation; (2) first-strand cDNA synthesis employing random hexamer primers and M-MuLV Reverse Transcriptase (RNase H-deficient); (3) second-strand cDNA generation using DNA Polymerase I with RNase H treatment; (4) blunt-end conversion of DNA fragments through coordinated exonuclease/polymerase activity; (5) adenylation of 3’ termini for adapter ligation using hairpin-loop structured adapters. Library fragments were size-selected (370–420 bp) through dual-stage AMPure XP bead purification (Beckman Coulter, Newton, MA, USA). Amplification was performed using the ABI 2720 Thermal Cycler (Applied Biosystems, Foster City, CA, USA) with Phusion High-Fidelity DNA Polymerase, Universal PCR primers, and index-specific primers. Final library validation was conducted via the Bioanalyzer 2100 system. RNA sequencing data in BAM format have been deposited in the NCBI SRA database under accession number SUB15236036.

### 2.4. Clustering and Quality Control of RNA-seq Libraries

Clustering generation of the index-coded samples was performed on a cBot Cluster Generation System using TruSeq PE Cluster Kit v3-cBot-HS (Illumia, CA, USA) and according to the manufacturer protocols. Post-clustering, sequencing libraries were processed through the Illumina Novaseq 6000 platform to generate 150 bp paired-end reads.

FASTQ-formatted raw reads underwent primary processing through fastp software v0.19.7 (Parameter: fastp -g -q 5 -u 50 -n 15 -l 150) for quality-controlled datasets. This workflow sequentially removed (1) adapter-contaminated reads, (2) poly-N sequences (>10% ambiguous bases), and (3) low-quality fragments (Phred score < 20 across > 50% of bases). Concurrent quality metric quantification included Q20 (base call accuracy ≥ 99%), Q30 (accuracy ≥ 99.9%), and GC content analysis. All subsequent analyses utilized these rigorously filtered datasets.

### 2.5. Analysis of Sequencing Data

Reference genome and gene model annotation files (OryCun2.0) were retrieved from the Ensembl genome database (OryCun2.0, Ensembl Release 110, Published in 2023). Paired-end clean reads were aligned to the reference genome by using Hisat 2.0.5. FPKM (Fragments Per Kilobase of transcript sequence per million base pairs sequenced) was calculated to quantify gene expression level. Reads counts mapped to each gene were generated through FeatureCounts v1.5.0-p3, followed by FPKM calculation based on the length of the gene and corresponding mapped reads.

Differential expression analysis of two breeds in the same muscle tissue was performed by using the DESeq2 R package v1.20.0 (*p*-value < 0.05,|log_2_FoldChange| ≥ 0.0 and |log_2_FoldChange| ≥ 1.0). Gene Ontology (GO) enrichment analysis of DEGs was performed using clusterProfiler R package 3.8.1. (padj < 0.05), and KEGG pathway analysis and protein–protein interaction (PPI) network construction were implemented through STRING database v10.5 (http://string-db.org/, accessed on 30 January 2025) [27]. Biological network visualization and analysis were conducted using Cytoscape v3.7.0 (http://cytoscape.org/, accessed on 30 January 2025). Network nodes were designated as “hub genes” based on their high degree of protein–protein interactions.

## 3. Results

### 3.1. The Changes in the Area of Muscle Fibers of Rabbits’ Skeletal Muscle at Different Stages

Ju rabbits exhibited significantly higher body weight compared to Ma rabbits from 3 weeks of age onward (Figure 1A). To characterize development change in thigh muscle and *longissimus dorsi* muscle morphology, paraffin sections of two muscle tissue across three developmental stages (Figure 1B,C). Histological analysis revealed significant age-dependent increases in muscle fiber cross-sectional area (CSA). At 56 days postnatally, Ju rabbits demonstrated significantly greater CSA than Ma rabbits in both muscle types (Figure 1D,E, Appendix A), though no interbreed differences were observed in total muscle area for either thigh or *longissimus dorsi* (Figure 1F, Appendix A). Subsequently, transcriptomic profiles compared gene expression patterns between the two breeds in both muscle tissues at the 56-day developmental stage.

### 3.2. Summarization of RNA-seq

Twelve RNA-seq libraries were generated from thigh and *longissimus dorsi* muscle at 56 days. All libraries produced raw sequencing data ranging from 40.25 to 47.07 million reads. Following stringent quality filtering, the libraries retained 6.51 Gb of high-quality clean bases, representing 98.01% of the original raw reads. All libraries demonstrated high sequencing quality, with Q30 scores averaging 94.58% (Table 2).

Transcriptomic characterization revealed distinct mRNA compositional patterns, with candidate transcripts predominantly originating from exonic regions (77.33%), followed by intronic (7.93%) and intergenic sequences (14.74%) (Figure 2A, Appendix A). Gene expression quantification using FPKM values demonstrated consistent transcriptional profiles across biological replicates, as visualized through boxplot distributions (Figure 2B).

Co-expression analysis revealed significant pairwise correlations among target genes (Figure 2C). Comparative analysis demonstrated stronger concordance between technical replicates within identical tissue–breed combinations (r = 0.950, *p* < 0.01) than intra-breed comparisons (r = 0.942, *p* < 0.01). Similarly, intra-breed correlations (r = 0.945, *p* < 0.01) and intra-tissue associations (r = 0.937, *p* < 0.01) exceeded cross-breed–tissue correlations (r = 0.923, *p* < 0.01). Hierarchical clustering of DEGs resolved four distinct expression clusters corresponding to breed–tissue combinations: Ju_Leg, Ju_Long, Ma_Leg and Ma_Long (Figure 2D). These findings collectively confirm the high reproducibility and analytical reliability of the RNA-seq data.

### 3.3. Differential Expression Analysis of mRNAs in Rabbit Skeletal Muscle

Venn diagram analysis identified 11,492 and 11,728 co-expressed genes in the thigh and *longissimus dorsi* muscles, respectively, across both breeds (Figure 3A,B). Comparative analysis revealed 1,983 DEGs in thigh muscle (785 upregulated and 1,198 downregulated), and 1,955 DEGs in *longissimus dorsi* muscle (734 upregulated and 1,221 downregulated) between Ma and Ju breeds using the thresholds (|log_2_FC| ≥ 0.0 and padj < 0.05; (Figure 3C, Appendix A). Application of stricter thresholds (|log_2_FC| ≥ 1 and padj < 0.05) resolved 284 DEGs in thigh muscle (90 upregulated, 194 downregulated) and 305 DEGs in *longissimus dorsi* muscle (106 upregulated and 199 downregulated) were identified (Figure 3D).

### 3.4. GO Enrichment and KEGG Pathway of DEGs in Thigh and Longissimus Dorsi Muscle

Functional characterization focused on DEGs identified through lenient screening thresholds (|log_2_FC| ≥ 0.0, padj< 0.05) between Ma and Ju. In thigh muscle, Gene Ontology (GO) enrichment analysis demonstrated significant association with 706 biological process terms, 41 cellular component terms, and 62 molecular function terms (Appendix A). Concurrent KEGG pathway analysis revealed 64 significantly enriched pathways (*p* < 0.05). Both analyses identified functional terms mechanistically linked to myogenic processes (Table 3).

In *longissimus dorsi* muscles, GO enrichment analysis demonstrated a significant association of DEGs with 500 biological process terms, 52 cellular component terms, and 49 molecular function terms (Appendix A). KEGG pathway analysis identified 64 significantly enriched signaling pathways (*p* < 0.05). Both analyses revealed functional annotations mechanistically linked to skeletal muscle development (Table 4).

Comparative analysis of thigh and *longissimus dorsi* muscle revealed cross-tissue characteristics in DEG functional enrichment, with three shared GO term (muscle cell migration, smooth muscle cell migration and myofibril) and two overlapping KEGG pathways (HIF-1 signaling pathway and Glycolysis/Gluconeogenesis). These conserved functional signatures indicate that the HIF-1 signaling pathway and Glycolytic regulation are mechanistically associated with interbreed variation in muscular development among rabbit lineages.

### 3.5. GO Enrichment, KEGG Pathway and PPI of Top 50 DEGs

Secondary analysis employed stringent selection criteria (|log_2_FC| ≥ 1 and padj < 0.05) to identify high-confidence DEGs between Ma and Ju breeds. These candidate DEGs underwent GO enrichment analysis, KEGG pathway analysis and protein–protein interaction (PPI) network evaluation.

In thigh muscle, these DEGs were significantly enriched in two principal domains: myogenic development and molecular interaction mechanisms. GO analysis revealed pronounced enrichment (Figure 4A, Appendix A) for muscle morphogenesis terms including muscle cell migration (GO:0014812), positive regulation of cell proliferation (GO:0008284), and smooth muscle cell migration (GO:0014909), alongside molecular binding processes such as regulation of cell adhesion (GO:0030155), cell adhesion molecule binding (GO:0050839), and extracellular matrix interactions (fibronectin binding GO:0001968; integrin binding GO:0005178). KEGG pathway analysis identified four significantly enriched signaling cascades (Figure 4B, Appendix A): cancer-associated MicroRNAs (ocu05206), PI3K-Akt signaling pathway (ocu04151), HIF-1 signaling pathway (ocu04066) and ECM-receptor interaction (ocu04512). PPI network analysis resolved two functionally distinct modules (Figure 4C, Appendix A), with PRL emerging as the central hub gene in Cluster 1, while Cluster 2 contained multiple regulatory nodes including NF-κB, BCL, MYC, and GAPDH.

In *longissimus dorsi* muscle, these DEGs exhibited significant functional enrichment spanning biological processes associated with myogenic developmental processes, including muscle organ morphogenesis (GO:0048644), muscle structure development (GO:0061061), and muscle tissue morphogenesis (GO:0055008) (Figure 4D, Appendix A). Cellular component analysis revealed pronounced enrichment in contractile fiber structures (GO:0043292, GO:0030016) and cytoskeletal assemblies (contractile actin filament bundle GO:0097517; actin filament bundle GO:0032432).

KEGG pathway analysis identified ten significantly enriched pathways (Figure 4E, Appendix A), including cardiomyopathy-associated signaling (dilated cardiomyopathy (ocu05414), hypertrophic cardiomyopathy (ocu05410)), contractile regulation (cardiac muscle contraction (ocu04260), metabolic processes (glycolysis/gluconeogenesis (ocu00010)), and oncogenic pathways (proteoglycans in cancer (ocu05205)). Protein–protein interaction (PPI) network analysis resolved three molecular clusters (Figure 4F, Appendix A), with PRL serving as the central hub in Cluster 1, GAPDH as the primary node in Cluster 2, and MYL/MYH family members dominating Cluster 3. Collectively, these findings demonstrate breed-specific DEG functional specialization and highlight intermuscular functional divergence between thigh and *longissimus dorsi* muscle.

### 3.6. The Analysis of Co-Expressive and Tissue-Specific Expressive DEGs in Thigh and Longissimus Dorsi Muscle

Finally, integrated analysis of co-expressive and tissue-specific expressive DEGs identified 124 conserved transcriptional regulators across thigh and *longissimus dorsi* muscle (Figure 5A). These shared DEGs demonstrated significant functional enrichment in Golgi-vacuolar transport mechanisms (GO:0006896) and membrane trafficking complexes (AP-3 adaptor complex (GO:0030123), AP-type membrane coat adaptor complex (GO:0030119)) (Figure 5D, Appendix A). Molecular characterization revealed dual functional specialization: 1) cellular component organization (membrane coat (GO:0030117), coated membrane (GO:0048475)), 2) metabolic catalysis (NAD(P)-dependent oxidoreductase activity (GO:0016620, GO:0050661), NAD binding (GO:0051287)). PPI network analysis positioned GAPDH as the central regulatory hub coordinating these conserved DEGs (Figure 5E). These findings suggest that interbreed differences in glucose-mediated energy metabolism may underlie observed muscular developmental disparities between Ma and Ju rabbits.

The *longissimus dorsi* muscle-specific DEGs showed significantly enrichment in GO terms (Figure 5B, Appendix A) associated with purine and nucleoside biosynthetic processes, including ATP biosynthesis (GO:0006754), purine ribonucleoside triphosphate biosynthesis (GO:0009206), and proton transport (energy-coupled proton transport (GO:0015985), ATP synthesis-coupled proton transport (GO:0015986), mitochondrial proton-transporting ATP synthase complexes (GO:0005753, GO:0045259)). Protein–protein interaction (PPI) analysis revealed these DEGs clustered into three distinct networks (Figure 5C), with SOX, MYH/MYL, and PRL identified as key hub genes.

Thigh muscle-specific DEGs demonstrated significant functional enrichment across multiple biological domains (Figure 5F, Appendix A), including structural organization (structural constituent of ribosome (GO:0003735)), immune modulation (chemokine activity (GO:0008009), chemokine receptor binding (GO:0042379)), cytokine interactions (cytokine activity (GO:0005125), cytokine receptor binding GO:0005126)) and Molecular recognition (G protein-coupled receptor binding (GO:0001664), Hsp70 protein binding (GO:0030544), Unfolded protein binding (GO:0051082)) and Metabolic catalysis (NAD(P)-dependent oxidoreductase activities (GO:0016620, GO:0016903), NAD binding (GO:0051287)). These DEGs formed a core regulatory network consisting of NF-κB, BCL, CXCL10 and PRL to govern muscle-specific transcriptional programs.

## 4. Discussion

Skeletal muscle development constitutes a multistage morphogenetic process, progressing through myogenic differentiation from mesoderm-derived precursors to myoblasts, subsequent fusion into multinucleated myotubes, and terminal maturation into functional muscle fibers [28]. While skeletal muscle morphology has been extensively characterized in porcine [29,30], avian [31], and ovine [32] models, lagomorph skeletal myogenesis remains understudied.

Histomorphometric analysis of fetal, juvenile, and adult rabbit hindlimb musculature through paraffin sectioning revealed statistically significant (*p* < 0.01) ontogenetic variations in muscle fiber diameter, numerical density, cross-sectional area, and population density across developmental stages [33]. Notably, fetal specimens at 2 weeks’ gestational age displayed incomplete myofiber maturation, with histological examination showing sparse primary myocyte populations [34]. Comparative analyses indicate complete secondary myofiber formation accompanied by significant diameter expansion (p < 0.05) in juvenile and adult specimens. These structural investigations delineate stage-specific myoarchitectural transformations in leporine hindlimb muscles, establishing a foundational framework for understanding lagomorph skeletal muscle developmental biology.

The advent of high-throughput sequencing technologies has revolutionized systematic investigations of growth-regulatory genes in animal development, generating expansive transcriptional datasets across mammalian species. Transcriptomic profiling of Jinghai yellow chicken breast muscle through RNA sequencing (RNA-Seq) across developmental stages identified RAC2 as a potential regulator of cellular proliferation via PAKs/MAPK8 pathway modulation. Comparative RNA-Seq analysis of skeletal muscle in crossbred beef cattle revealed substantial inter-cohort variation in differentially expressed genes (DEGs), suggesting breed-specific and environmental influences on feed efficiency-associated transcriptional networks.

To investigate molecular pathways governing skeletal muscle development, we conducted RNA sequencing (RNA-Seq) on thigh and longissimus dorsi muscle tissues from Sichuan Ma and Checkered Ju rabbits at 56 days. Differential expression analysis identified 1,983 DEGs in thigh muscle (785 upregulated; 1,198 downregulated) between breeds, with functional enrichment revealing critical biological processes: Myogenic regulation (muscle cell migration, smooth muscle cell migration), cellular homeostasis (negative regulation of cell death), structural organization (sarcomere assembly, myofibril formation, actin filament bundling) and metabolic coordination (organic acid binding, calmodulin binding, NADP binding). KEGG pathway analysis demonstrated enrichment in energy metabolism (Glycolysis/Gluconeogenesis) and mechanotransduction signaling (HIF-1, PI3K-Akt, FoxO, and MAPK pathways)

The glycolytic pathway facilitates myoblast fusion through ATP provision during embryonic myogenesis [35]. PI3K-Akt signaling promotes muscle hypertrophy via coordinated regulation of protein synthesis and atrophy suppression [36]. FoxO transcription factors modulate atrophy-related gene networks, where their inhibition prevents myofiber degeneration [37]. The p38 MAPK cascade orchestrates myogenic differentiation through transcriptional regulation of muscle-specific genes [38].

Comparative transcriptomic analysis revealed significant differential expression patterns between Ma and Ju leg muscles, with upregulated genes (GAPDH, PGM1, IGF1, SGK1) and downregulated ACSS1 demonstrating potential regulatory roles in leporine myogenesis. Mechanistic studies indicate these genes coordinate critical myogenic processes. In the processes of metabolic regulation, GAPDH (glyceraldehyde-3-phosphate dehydrogenase) exhibits age-dependent expression declines in avian pectoral muscle, correlating with reduced DETs during maturation [39]. In the processes of myoblast modulation, PGM1 (phosphoglucomutase 1) governs myoblast proliferation and inosine monophosphate deposition in poultry models [40]. In the processes of growth signaling cascade, IGF1 (insulin-like growth factor 1) mediates protein synthesis and hypertrophic growth across vertebrate musculature [41]. In the processes of proteostasis control, SGK1 (serum/glucocorticoid-regulated kinase 1) maintains muscle mass through coordinated suppression of autophagy and proteolysis while enhancing anabolism [42]. In the processes of mitochondrial dynamics regulation, ACSS1 (acyl-CoA synthetase short-chain 1) inversely correlates with gastrocnemius mass, suggesting regulatory involvement in muscle atrophy pathways [43]. These collective findings suggest conserved functional conservation of myoregulatory networks across mammalian species, with identified DEGs constituting key molecular determinants of lagomorph skeletal muscle development.

KEGG pathway analysis highlighted enrichment in hypoxia adaptation (HIF-1 signaling), energy metabolism (Glycolysis/Gluconeogenesis), mechanotransduction (PPAR signaling, actin cytoskeleton regulation), and PPAR-β activation enhances muscle regeneration through immunomodulation and transcriptional upregulation of myogenic factors (Pax7, MyoD, Myf5, Myogenin) [44]. Differential expression profiling identified elevated levels of GAPDH (glycolytic flux), ALDOA (glycogenolysis regulation), ACTN1 (α-actinin-1, cytoskeletal anchoring), and MYH10 (myosin heavy chain 10) in Ma versus Ju breeds, contrasted by suppressed ACSS1 expression. Functional annotations of these DEGs align with established roles in some previous evidence; ALDOA mediates glycolytic pathway coordination [45]. ACTN1 is indispensable for neuromuscular junction development [46]. MYH10 epigenetic regulation correlates with environmental adaptation in marine teleosts [47].

Functional enrichment analysis of the top 50 DEGs identified critical biological pathways associated with leporine myogenesis. In thigh muscle, significant GO terms included myogenic regulation (muscle cell migration, positive regulation of cellular proliferation, smooth muscle cell migration), signaling pathways (PI3K-Akt signaling, HIF-1 signaling, ECM-receptor interaction).

Protein–protein interaction (PPI) networks revealed PRL, NF-κB, BCL, MYC, and GAPDH as central regulatory hubs. The extracellular matrix (ECM) serves as a biochemical niche facilitating satellite cell activation through cytokine release (FGF2, HGF, SDF-1), which induces myogenic regulatory factors (MyoD, Myf5, MyoG) during muscle regeneration [48]. Some evidence demonstrates that PRL enhances cold-stress-induced myoblast proliferation in avian models [49], NF-κB modulates skeletal muscle metabolic homeostasis and mass maintenance [50], BCL-2 regulates myogenic progenitor clonal expansion [51], c-MYC inhibits differentiation while promoting myoblast proliferation/hypertrophy [52].

In longissimus dorsi muscle, enriched GO terms encompassed structural organization (contractile fiber assembly (myofibril, stress fiber), actin cytoskeletal bundling), developmental processes (muscle organ morphogenesis, tissue patterning). Significant KEGG pathways included hypoxia response (HIF-1), energy metabolism (glycolysis/gluconeogenesis), and ribosomal biogenesis. PPI analysis identified PRL, GAPDH, and MYL/MYH as nodal regulators, with mechanistic studies showing MYH-embryonic isoform directs mammalian myoblast differentiation [53], and MYL4 modulates porcine muscle fiber type specification and growth kinetics [54].

## 5. Conclusions

This investigation employed RNA sequencing (RNA-Seq) to comparatively analyze transcriptomic profiles of thigh and longissimus dorsi muscles between Sichuan Ma and Checkered Ju rabbits. Analytical results revealed differentially expressed genes (DEGs) predominantly enriched in critical metabolic and signaling pathways, particularly Glycolysis/Gluconeogenesis and HIF-1 signaling pathway, which are mechanistically associated with myogenic development. These findings provide crucial mechanistic insights for elucidating muscular regulatory networks in lagomorphs, while establishing foundational data for optimizing molecular breeding strategies targeting Ma rabbit muscle phenotypes.

## Figures and Tables

**Figure 1 animals-15-01585-f001:**
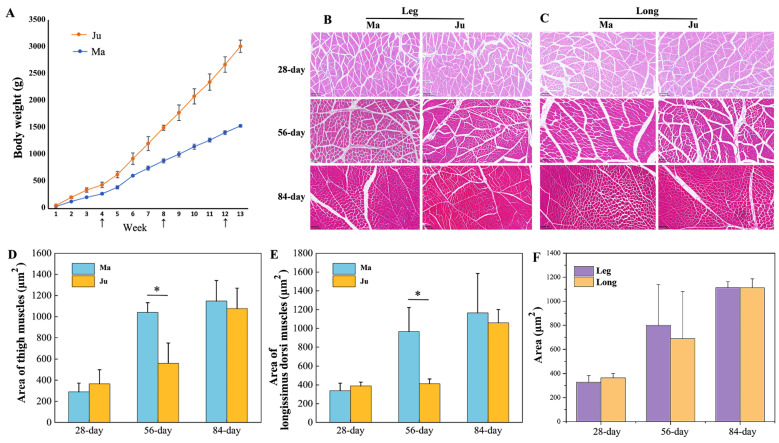
The areas of muscle fibers at different stages. (**A**) The change in body weight of Ma and Ju rabbit from 1-week to 13-week age. (**B**) The H&E staining of thigh muscles. (**C**) The H&E staining of *longissimus dorsi* muscles. (**D**) The area of thigh muscle at 28, 56, 84 days. (**E**) The area of *longissimus dorsi* muscles at 28, 56, 84 days (* *p* < 0.05). (**F**) The comparison of muscle fiber area between thigh and *longissimus dorsi* muscles.

**Figure 2 animals-15-01585-f002:**
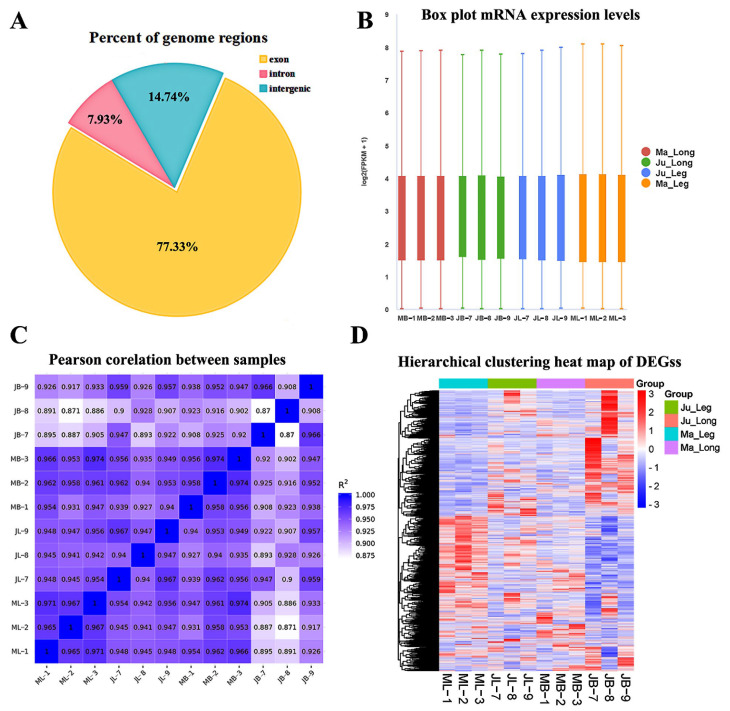
Summarization of RNA-seq data. (**A**) Distribution of gene expression levels in different samples. (**B**) The distribution of FPKM in all samples. (**C**) The correlation analysis of all samples. (**D**) Hierarchical clustering heat map of all DEGs. ML: thigh muscle of Ma rabbit; MB: *longissimus dorsi* muscles of Ma rabbit; JL: thigh muscle of Ju rabbit; JB: *longissimus dorsi* muscles of Ju rabbit.

**Figure 3 animals-15-01585-f003:**
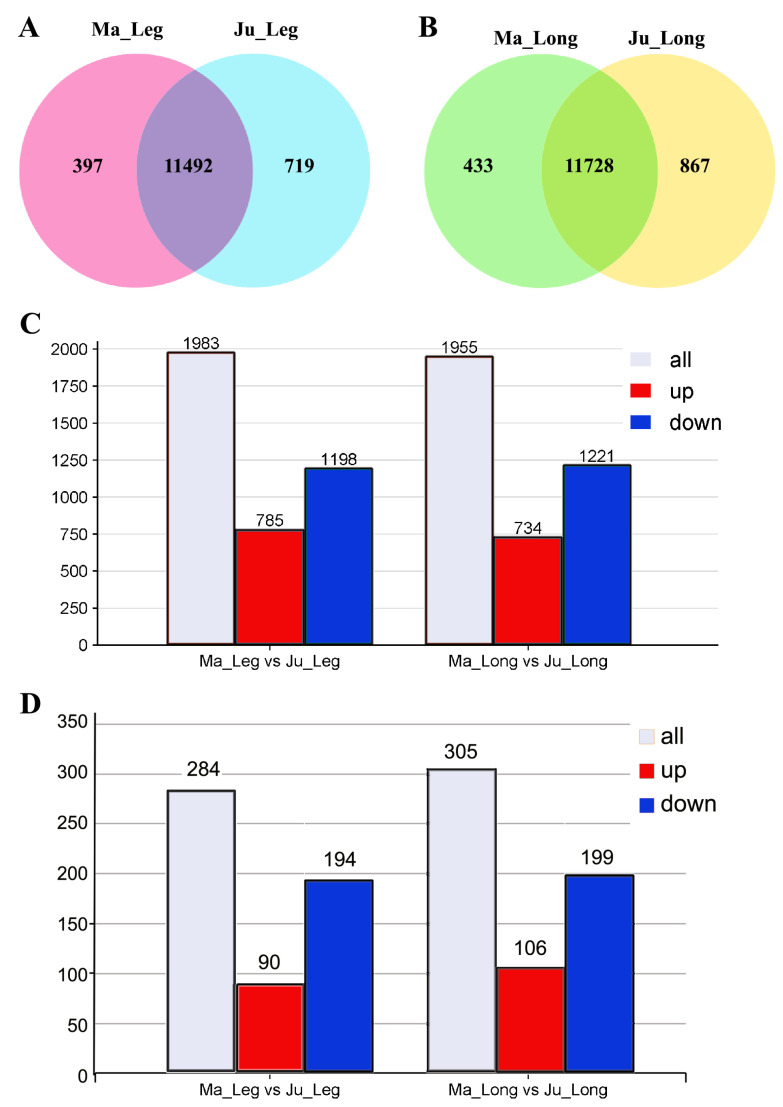
Differentially expressed genes in rabbit muscle tissues. (**A**) Venn diagram of co-expressive gene in the thigh muscle of Ma and Ju rabbit. (**B**) DEGs of Ma vs. Ju rabbit using thresholds |log_2_FC| > 0. (**C**) DEGs of Ma vs. Ju rabbit using |log_2_FC| > 0 method. (**D**) DEGs of Ma vs. Ju rabbit using |log2FC| > 1. Ma_Leg: Thigh muscle from Ma rabbit; Ju_Leg: Thigh muscle from Ju rabbit; Ma_Long: *Longissimus dorsi* muscle from Ma rabbit; Ju_Long: *Longissimus dorsi* muscle from Ju rabbit.

**Figure 4 animals-15-01585-f004:**
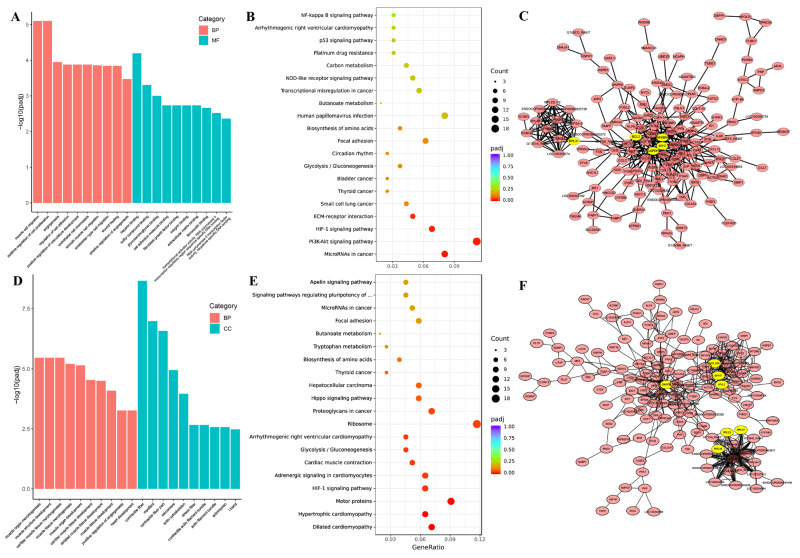
GO enrichment, KEGG pathway and PPI of DEGs in thigh and *longissimus dorsi* muscle. GO enrichment (**A**), KEGG (**B**) and PPI (**C**) analysis of DEGs in the thigh muscle; GO enrichment (**D**), KEGG (**E**) and PPI (**F**) analysis of DEGs in the *longissimus dorsi* muscle.

**Figure 5 animals-15-01585-f005:**
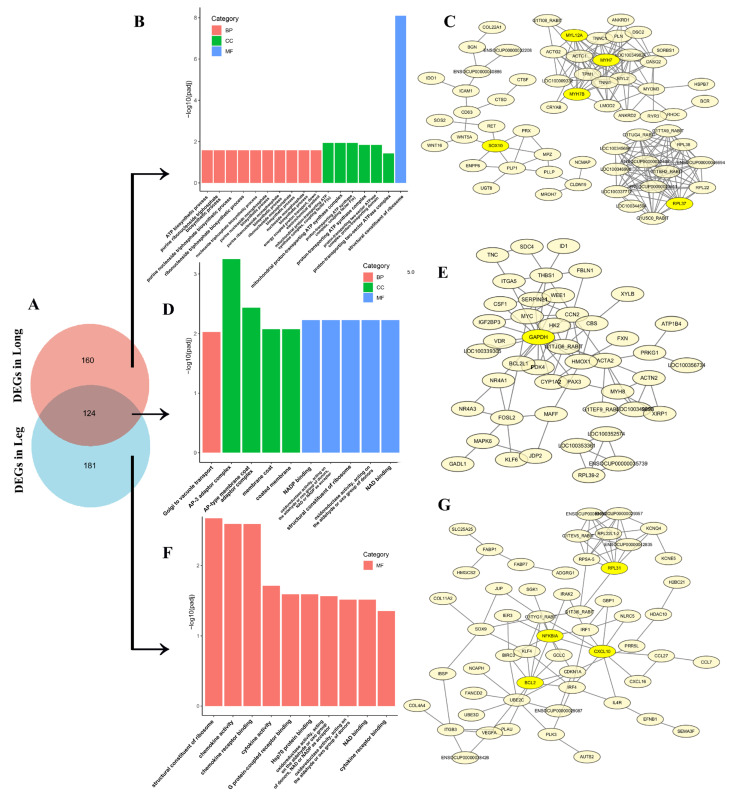
Co-expressive and tissue-specific DEGs in thigh and *longissimus dorsi* muscle. (**A**) The Venn diagram of DEGs (Ma vs. Ju) in thigh and *longissimus dorsi* muscle; (**B**) GO analysis of the *longissimus dorsi* muscle-specific DEGs; (**C**) PPI of the *longissimus dorsi* muscle-specific DEGs; (**D**) GO analysis of co-expressive DEGs in thigh and *longissimus dorsi* muscle; (**E**) PPI of co-expressive DEGs in thigh and *longissimus dorsi* muscle; (**F**) GO analysis of the thigh muscle-specific DEGs; (**G**) PPI of the thigh muscle-specific DEGs.

**Table 1 animals-15-01585-t001:** Ingredient composition of experimental diets (%, as feed basis).

Ingredient	Content (%)
Alfalfa meal	34.4
Corn	21.5
Soya bean meal	13.5
Bran	22.5
Wheat	5
Calcium hydrogen phosphate	0.9
Mountain flour	0.7
NaCl	0.5
The vitamin premix ^1^	0.5
The mineral Premix ^2^	0.5
Total	100.00
**Nutrient Level ^3^**	**Content**
Digestible energy, Mcal/kg	10.45
Crude protein, %	17.07
Crude fibre, %	10.92
Lysine, %	0.84
Methionine + Cystine, %	0.62
Calcium, %	1.01
Phosphorus, %	0.65
Neutral detergent fiber, %	27.22
Acid detergent fiber, %	14.29

^1^ The vitamin premix provides the following per kilogram of diet: VA 18000 IU; VD3 6000 IU; VE 48 IU; VK3 6 mg; VB1 6 mg; VB2 15 mg; VB6 7.2 mg; VB12 720 μg; D-pantothenic acid 30 mg; nicotinic acid 60 mg; folic acid 3 mg; biotin 3 mg. ^2^ The mineral premix provides the following per kilogram of diet: copper (CuSO_4_.5H_2_O) 6 mg; iron (FeSO_4_.H_2_O) 100 mg; zinc (ZnSO_4_.H_2_O) 100 mg; manganese (MnSO_4_.H_2_O) 4 mg; iodine (KI) 0.14 mg; selenium (Na_2_SeO3) 0.3 mg. ^3^ Nutrient levels are calculated values.

**Table 2 animals-15-01585-t002:** Sample sequencing data quality summary and reference genome comparison.

Sample	Clean Reads	Clean Bases	Mapped Ratio	Uni-Mapped	Q30	GC (%)
Ma_Leg_1	42394110	6.36G	88.86%	84.39%	94.47	54.55
Ma_Leg_2	41266572	6.19G	90.21%	85.58%	93.77	53.29
Ma_Leg_3	42051878	6.31G	89.78%	85.37%	94.02	53.67
Ju_Leg_1	42406862	6.36G	87.94%	83.85%	94.08	55.13
Ju_Leg_2	41415140	6.21G	89.51%	85.25%	95.2	54.2
Ju_Leg_3	41681174	6.25G	89.35%	84.76%	94.13	53.93
Ma_Long_1	46907154	7.04G	84.05%	79.4%	94.66	54.42
Ma_Long_2	43304158	6.5G	86.49%	82.52%	93.74	55.95
Ma_Long_3	39350026	5.9G	87.75%	83.68%	93.9	55.46
Ju_Long_1	44311234	6.65G	87.66%	83.11%	95.35	54.63
Ju_Long_2	42472108	6.37G	82.41%	78.17%	95.71	57.76
Ju_Long_3	42970742	6.45G	87.9%	83.47%	95.88	55.27
average	42544263.17	6.51G	87.66%	83.30%	94.58	54.86

**Table 3 animals-15-01585-t003:** GO and KEGG of DEGs related to muscle development in thigh muscle.

Category	GOID	Description
BP	GO:0014812	muscle cell migration
BP	GO:0043069	negative regulation of programmed cell death
BP	GO:0014909	smooth muscle cell migration
CC	GO:0030017	sarcomere
CC	GO:0030016	myofibril
CC	GO:0032432	actin filament bundle
MF	GO:0043177	organic acid binding
MF	GO:0005516	calmodulin binding
MF	GO:0050661	NADP binding
MF	GO:0005509	calcium ion binding
KEGG	KEGGID	Description
	ocu00010	Glycolysis/Gluconeogenesis
	ocu04066	HIF-1 signaling pathway
	ocu04151	PI3K-Akt signaling pathway
	ocu04068	FoxO signaling pathway
	ocu04010	MAPK signaling pathway

**Table 4 animals-15-01585-t004:** GO and KEGG of DEGs related to muscle development in *longissimus dorsi* muscles.

Category	GOID	Description
BP	GO:0014812	muscle cell migration
BP	GO:0006954	inflammatory response
BP	GO:0014909	smooth muscle cell migration
BP	GO:0061061	muscle structure development
CC	GO:0015629	actin cytoskeleton
CC	GO:0043292	contractile fiber
CC	GO:0030016	myofibril
CC	GO:0016459	myosin complex
MF	GO:0005509	calcium ion binding
MF	GO:0001968	fibronectin binding
MF	GO:0051015	actin filament binding
KEGG	KEGGID	Description
	ocu04066	HIF-1 signaling pathway
	ocu00010	Glycolysis/Gluconeogenesis
	ocu04810	Regulation of actin cytoskeleton
	ocu03320	PPAR signaling pathway

## Data Availability

The transcriptomic data have been stored in the SRA database of NCBI: https://dataview.ncbi.nlm.nih.gov/object/PRJNA1254676, accessed on 30 January 2025 (Acession No.: SRR33301172; SRR33301173; SRR33301174; SRR33301175; SRR33301166; SRR33301170; SRR33301171; SRR33301168; SRR33301167; SRR33301169; SRR33301164; SRR33301165).

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
