# Peer review of "Comparative Transcriptomic Analysis Between High- and Low-Growth-Rate Meat-Type Rabbits Reveals Key Pathways Associated with Muscle Development"

_animals, 2025, doi:10.3390/ani15111585_

Round 1
Reviewer 1 Report
Comments and Suggestions for Authors
The comparison transcriptome between high and low growth ratio of meat-type rabbit revealed the primary pathway related 3 to muscle development
Summary of the study
This paper examines the biological processes that mechanistically explain muscle growth in rabbits. To achieve this objective, they used local rabbit breeds with known divergent morphological traits, namely the agile Sichuan linen (Ma) rabbit and the huge-bodied Checkered Giant (Ju) rabbit. The investigators harvested muscle from the thigh region and longissimus dorsi muscles and subsequently subjected them to histological and RNA-sequencing. They ran downstream analysis on the DEGs and identified numerous biological processes which, they speculated, played key roles in muscle development. Overall, the paper is written in subpar language, and the grammar is wanting. The introduction lacks pertinent information regarding the reasons for the choice of the animal model and the two muscles. The method section lacks specifics of samples harvested, processing and analysis, which would allow for the repetition of the study by someone else. No data on the validation of DEGs is provided. The result section shows double work from the identification of DEGs to downstream analysis, resulting in a whole lot of biological processes. The discussion section lacks pertinent literature with rigorous depth that would have supported the results and the author’s arguments. As such, this paper would benefit from a thorough slimdown of the DEG analysis and a more focused approach on the discussion of the results and conclusion. Further, the authors need to avoid copying word for word from other published literature as this diminishes the efforts of independent scientific writing.
Detailed review
Introduction
- Line 53 need to italicize the name “Oryctolagus cuniculus”. The same should be done for any other scientific name.
- Line 53 to 62 can be considered plagiarism since this whole portion of the introduction has been copied, exactly as is, from the introduction section of a previously published paper (ref 31) in the reference section (line 514).
- What are the specific reasons why the thigh and longissimus dorsi types were chosen for the study? The reason provided in line 57 is too shallow and lacks references.
- Line 56: What do the authors mean by “sensitization”?
- Line 82: What do the authors mean here by “Muscle transcriptional helper studies”?
Materials and methods
- The thigh muscle as a whole comprises different muscle groups. What is the specific name (eg, sartorius, quadriceps, etc) of the thigh muscle group did the investigators harvest and studied? This is very important. If possible, provide the anatomical rendering (eg, a picture/drawing) of the specific sites where the muscles were harvested, and indicate whether this was consistent for all samples. Further, what leg was the muscle harvested from? What side was the longissimus dorsi muscle harvested from? This information should be provided at the first instance. Thereafter, you can continue mentioning thigh muscle without being specific with the name of the muscle.
The authors could consider reading this publication: J. Anat. (2000) 196, pp., 203–209, Histomorphology of rabbit thigh muscles: establishment of standard control values
- What was the size/measurement of the muscle samples harvested and processed for each group?
- Is Sakura a type of muscle-specific fixation solution under line 98? Provide details of the contents.
- Line 117-131: How was total RNA extracted, including the kits used? Were there any kits used to enrich for mRNA?
- Line 137: What was the version of FASTP software used?
- What type of reference genome did they use? Indicate the version and release year. The same is required for the annotation file.
Line 174: Change nest to next
Line 173 and figure F.: How were the measurements for both muscles obtained? Need an explanation.
Line 181: The E should be F. Currently, there are 2 Es.
Line 224-228: The legend in Figure 3 is confusing and does not match the figure.
Did the authors consider weighing individual thigh and longissimus muscles for each breed and then compare?
Figure 1 legend. The legend indicates diameter, while the text and figure show muscle areas.
Have the investigators determined the muscle fiber-type of the two muscles studied? This information is out in the literature, but it has not been captured in the paper. The muscle fiber-type directly informs its primary form of metabolism, especially energy metabolism.
What is the role of body weight and muscle cross-sectional area in muscle growth between the two breeds? So far, these measurements are antagonistic. What are the authors' explanations for this?
What was the primary objective of analyzing the DEGs twice: first, using the low threshold and then the high threshold? This results in unnecessary duplicity, biases, misinterpretation, and confusion. The ideal situation would have been to use the most stringent method that would generate unequivocally relevant DEGs and, subsequently, streamline your downstream analysis. As it is now, the paper is full of unnecessary biological terms and pathways that make it hard to understand and identify the real purpose of the study.
The whole paper has mentioned the statement “these DEGs may play an important role in muscle development" over and over without providing any critical literature to support their claim. Essentially, the authors did not provide sufficient literature-supported discussion on how specific pathways directly influence muscle development and growth. The literature cited is few or lacks depth sufficient to provide any meaningful support of the authors' arguments.
346: Change “measured” to “studied.”
Comments on the Quality of English LanguageThe quality of the English language is below par and needs a thorough review by an established entity.
Author Response
The summary of the study
Comment 1: Line 53 need to italicize the name “Oryctolagus cuniculus”. The same should be done for any other scientific name.
Response 1: we added the Latin name “Oryctolagus cuniculus” (Page 2,line 50)
Comment 2: Line 53 to 62 can be considered plagiarism since this whole portion of the introduction has been copied, exactly as is, from the introduction section of a previously published paper (ref 31) in the reference section (line 514).
Response 2: We have completely revised the introduction section and reduced the use of text in the literature..
As a lean and nutrient-dense protein source, rabbit meat has significant interest in agricultural animal production. In 2021, Global yield of rabbit meat was about 900,000 tons, from 570 million rabbits[1]. Moreover, rabbit have high reproductive efficiency (short gestation intervals, high parity rates) and feed conversion ratios.Legs are the main depots of skeletal muscle growth in rabbits[2].Thigh muscle is important part of food potential in rabbit meat[3]. Longissimus dorsi muscle is important indcator in meat quality and growth kinetics ,was studies due to its significant role in meat production and its relevance in muscle development studies for research [4]. Therefore, thigh and longissimus dorsi muscle is often used as research subjects in these studies related to muscle development, meat quality, fiber types.(Page 2,line50-58)
Comment 3:What are the specific reasons why the thigh and longissimus dorsi types were chosen for the study? The reason provided in line 57 is too shallow and lacks references.
Response 3: we explain the reason that is thigh and longissimus dorsi muscle.
As a lean and nutrient-dense protein source, rabbit meat has significant interest in agricultural animal production. In 2021, Global yield of rabbit meat was about 900,000 tons, from 570 million rabbits[1]. Moreover, rabbit have high reproductive efficiency (short gestation intervals, high parity rates) and feed conversion ratios.Legs are the main depots of skeletal muscle growth in rabbits[2].Thigh muscle is important part of food potential in rabbit meat[3].Longissimus dorsi muscle is important indcator in meat quality and growth kinetics ,was studies due to its significant role in meat production and its relevance in muscle development studies for research [4]. Therefore, thigh and longissimus dorsi muscle is often used as research subjects in these studies related to muscle development, meat quality, fiber types.
Comment 4:What do the authors mean by “sensitization”?
Response 4: In the article, it may be intended that rabbit meat is ‘hypoallergenic’, i.e., it is a safer food for some consumers, especially when compared to some other types of meat (e.g., beef, lamb) that may cause fewer allergic reactions.
Comment 5:What do the authors mean here by “Muscle transcriptional helper studies”?
Response 5:Changes have been completed in the paper.
In Materials and methods Section
Comment 1:The thigh muscle as a whole comprises different muscle groups. What is the specific name (eg, sartorius, quadriceps, etc) of the thigh muscle group did the investigators harvest and studied? This is very important. If possible, provide the anatomical rendering (eg, a picture/drawing) of the specific sites where the muscles were harvested, and indicate whether this was consistent for all samples. Further, what leg was the muscle harvested from? What side was the longissimus dorsi muscle harvested from? This information should be provided at the first instance. Thereafter, you can continue mentioning thigh muscle without being specific with the name of the muscle.
Response 1: We thank your comments. Actually, we ensure that the collected muscle are completely consistent among different individuals. How to collect the muscle, we added the detailed description in Materials and Methods.
Thigh muscle (the left quadriceps femoris was sampled at the midpoint between the greater trochanter and the lateral condyle of the femur about 0.5-1cm3) and longissimus dorsi muscle(the right longissimus dorsi was collected at the L3–L5 lumbar region , approximately 1.5 cm lateral to the vertebral column about 0.5-1cm3)
Comment 2:What was the size/measurement of the muscle samples harvested and processed for each group?
Response 2: Approximately 0.5-1 cubic centimeters of muscle mass.we added the detailed description in Materials and Methods.
Comment 3:How was total RNA extracted, including the kits used? Were there any kits used to enrich for mRNA?
Response 3: We added these Kit, and its Cat. No..
Total RNA was isolated from rabbit skeletal muscle (vastus lateralis and longissimus dorsi) using TRNzol Universal Reagent (Cat. No.:DP424, Tiangen Biotech, China).
Comment 5:What was the version of FASTP software used?
Response 5: Raw data (raw reads) of the fastq format were first processed through fastp software(0.19.7). fastp v0.19.7,parameters:fastp -g -q 5 -u 50 -n 15 -l 150.
Comment 6:What type of reference genome did they use? Indicate the version and release year. The same is required for the annotation file.
Response 6: We added the reference genmone version and it link to the manuscript. The Genome version:OryCun2.0:A chromosome-level assembly (primary assembly type: "toplevel").Release year: 2023 (via Ensembl Release 110, published in Q2 2023).Annotation file:Version: Corresponds to OryCun2.0.110 (matching Ensembl Release 110).Content: Gene models, transcripts, and functional annotations (GTF format).
Comment 7:Line 173 and figure F.: How were the measurements for both muscles obtained? Need an explanation.
Response 7: We attded the description in Materials and Methods.
Images of muscle tissues were captured using Evident APX100 (Olympus, Japan). The cross-sectional area (CSA) of muscle fiber was measured in 3 non-overlapping different eyefiled for each of sample using image J..
Comment 8:Line 224-228: The legend in Figure 3 is confusing and does not match the figure.
Response 8: We are sorry for this. It’s mistake. We have revised the figure legend, and then checked all figures and their figure lengends.
Comment 9:Did the authors consider weighing individual thigh and longissimus muscles for each breed and then compare?
Response 9: This is a omission. We didn’t weigh individual thigh and longissimus muscles at that time. I consider it taking long time to isolate the muscle,which lead to RNA degradion.
Comment 10:Figure 1 legend. The legend indicates diameter, while the text and figure show muscle areas.
Have the investigators determined the muscle fiber-type of the two muscles studied? This information is out in the literature, but it has not been captured in the paper. The muscle fiber-type directly informs its primary form of metabolism, especially energy metabolism.
What is the role of body weight and muscle cross-sectional area in muscle growth between the two breeds? So far, these measurements are antagonistic. What are the authors' explanations for this?
Response 10: We are sorry, it’s mistake, and it should be “area”. we revised it.
In the study, we only revealed these genes related muscle development. We'll considered to check the muscle fiber-type of the two muscles in our next studied.
The point of contradiction between body weight and area, we believe, is the reason for the difference in the number of muscle fibers.More muscle fibers, smaller muscle fiber area, but heavier weights.That's our guess, and we'll probably look into it next.

Reviewer 2 Report
Comments and Suggestions for Authors
The authors have submitted an article that outlines an interesting investigation of the comparison transcriptomic profiles related to muscle development between high and low growth ratio of two meat-type rabbit breeds. The results of this study are novel and could be important for scientists, rabbit producers and meat industry to improve meat quality and quantity.
The keywords accurately reflect the content of the manuscript. I suggest changing the title as outlined in the submitted PDF file. The authors of this manuscript gave us a clear introduction to the study based on the currently available scientific literature in this filed. Research aim of the study is clearly defined. The materials and methods are not clearly described and should be corrected as outlined in the PDF file. The obtained results are not well presented and should be corrected in accordance with appropriate statistical approach. Discussion section should be more direct and should focus more on obtained results. Despite the fact that my mother language in not English, manuscript should be checked by native speaker. References consist of appropriate and relevant papers.
I do believe this work is worthy of publication in Animals journal, but I would recommend a major changes before it is published. My comments and suggestions are outlined in the submitted PDF file.

Author Response
We would like to express our sincere gratitude to the reviewer for your insightful comments and valuable feedback.
Comment 1:My comments and suggestions are outlined in the submitted PDF file.
Response 1: We have revised the manuscript according to your suggestion in the manuscript. We list these revisions
- Rabbits were housed individually in single cages under standard husbandry conditionswith automatic water supply and manual feeding. Ma (n = 9) and Ju (n = 9) were feed with complete diet (Table 1) according to 3% body weight to supply dry matter intake daily. Thigh muscle (the left quadriceps femoris was sampled at the midpoint between the greater trochanter and the lateral condyle of the femur about 0.5-1cm3) and longissimus dorsi muscle(the right longissimus dorsi was collected at the L3–L5 lumbar region , approximately 1.5 cm lateral to the vertebral column about 0.5-1cm3).(Page2 line86-93)
- The first person in the paper has been changed
Child in the literatures means 6-week-old rabbits.(original text:including child stage rabbits at 6-week-old (0.86 ± 0.083 kg) and adult female rabbits with 2 weeks gestation at 6-month-old (4.37 ± 0.033 kg)).
Reviewer 3 Report
Comments and Suggestions for Authors
The manuscript focuses on an important topic in rabbit meat production by comparing gene expression between fast and slow-growing rabbit breeds. The study provides useful data, but the writing needs improvement in terms of grammar, clarity, and the explanation of results in terms of their biological significance.
Avoid using phrases like 'we did this' or 'we found that' throughout the manuscript. Scientific writing should be in the third person. Also, check the manuscript for proper tense usage and grammatical accuracy.
Throughout the text, please use standard formatting for gene and pathway names (e.g., IGF1, GAPDH instead of Igf1, Gapdh).
Title: The current title has grammatical issues. Suggested revision: "Comparative transcriptomic analysis between high and low growth rate meat-type rabbits reveals key pathways associated with muscle development."
Simple Summary: The summary should be simplified and use less technical jargon for better understanding by a general audience. For example, using terms like 'gene expression analysis' instead of 'transcriptome sequencing' would improve clarity. Additionally, correct 'differentiated expressed genes' to 'differentially expressed genes.
Abstract: The abstract should emphasize the main findings, highlighting their significance and relevance. Include the number of rabbits used in the study and ensure that abbreviations such as GO (Gene Ontology) and DEG (Differentially Expressed Genes) are clearly defined when first mentioned in the manuscript.
Introduction
Line 51: Please clarify the term 'meat rabbits'. Do you mean rabbits specifically raised for meat production?
Line 53: Italicize Oryctolagus cuniculus
Line 70–71: Please specify the geographical regions or provinces where Sichuan linen (Ma) and Checkered Giant (Ju) rabbits are commonly bred or originated. This will help readers understand the local relevance and breeding context of these two rabbit breeds.
Line 73: Consider rephrasing “leads to changes” → “leads to variations”
Line 77: Consider rephrasing “involved in regulating” → “that controls”
End the introduction with a clear research goal or question. Also, make sure the paragraph flows logically from general background to why these two rabbit breeds are being compared.
Materials and Methods: The Methods section should include sufficient detail, cite relevant references where appropriate, and specify the sources of materials and reagents used.
Line 90–106: Clearly define the two breeds (Ma and Ju) early on. Also, make sure to mention if the study had ethical approval. Please provide information on the housing and rearing conditions of the animals. Specify the RNA extraction method, including the name of the kit and manufacturer.
Line 115: Mention if you tested for normality and equal variance before using ANOVA.
Line 149: Mention the software used (like DESeq2) earlier and explain how you corrected for multiple comparisons (e.g., Benjamini-Hochberg method).
Results
Lines 166–174: There’s a mistake with reporting significance. You wrote "p > 0.05" which means not significant, but the text says there is a difference, please clarify.
Line 183: Mention how many reads mapped uniquely to the genome.
Line 210–228: No need to keep repeating the same thresholds (|log2FC| > 1, padj < 0.05). Explain them once clearly.
Line 229–247: Don’t just list the enriched pathways. Explain why they matter for muscle growth. How are glycolysis or HIF-1 involved?
Lines 250–281: Adding a new figure showing top GO terms for both muscles would help comparison.
Line 297: Briefly explain GAPDH's role both as a control gene and a functional gene in muscle metabolism.
Discussion: The discussion should be strengthened by reducing repetition of gene functions unless presenting new insights. Connect gene expression findings to specific rabbit traits, such as muscle size, and avoid speculative statements unless supported by data or literature.
Figures: Check figure labels (e.g., Figure 1 has two parts labeled “E”). Make sure all figures are clear and properly labeled with descriptive legends.
Comments on the Quality of English LanguageThe writing needs improvement for better grammar and clarity. Avoid using phrases like “we did” or “we found”. Scientific writing should be in third person. Also, make sure the tenses are used correctly throughout the manuscript.
Author Response
We would like to express our sincere gratitude to the reviewer for your insightful comments and valuable feedback.
Comment 1:Title: The current title has grammatical issues. Suggested revision: "Comparative transcriptomic analysis between high and low growth rate meat-type rabbits reveals key pathways associated with muscle development."
Response 1:Changes have been completed in the paper(Page1 line2).
Comment 2:Simple Summary: The summary should be simplified and use less technical jargon for better understanding by a general audience. For example, using terms like 'gene expression analysis' instead of 'transcriptome sequencing' would improve clarity. Additionally, correct 'differentiated expressed genes' to 'differentially expressed genes.
Response 2:Gene expression profiles of 56-day-old muscles identified thousands of differentially expressed genes linked to muscle growth.(Page1 line19-20)
Comment 3:Abstract: The abstract should emphasize the main findings, highlighting their significance and relevance. Include the number of rabbits used in the study and ensure that abbreviations such as GO (Gene Ontology) and DEG (Differentially Expressed Genes) are clearly defined when first mentioned in the manuscript.
Response 3: we selected 18 rabbits (9 Ma rabbits and 9 Ju rabbit) and first compared the body weight between two rabbit breeds with distinct growth rates.(Page1 line29-30)
Comment 4:Please clarify the term 'meat rabbits'. Do you mean rabbits specifically raised for meat production?
Response 4:Yes, the term "meat rabbits" refers specifically to rabbits that are bred and raised primarily for meat production. These rabbits are selectively bred for characteristics such as rapid growth rates, efficient feed conversion, and good meat yield, making them suitable for the commercial meat industry. In scientific literature, these animals are often referred to as "meat-type rabbits" or "commercial meat rabbits" to distinguish them from rabbits raised for other purposes, such as pets or show animals.
According to research, meat rabbits are typically of specific breeds or genetic lines that have been optimized for high productivity in terms of meat quality and quantity. For example, breeds such as the New Zealand White, Californian, and French Lop are commonly used in meat production due to their rapid growth rates and large body sizes. These rabbits are often raised in controlled environments to maximize production efficiency, which includes monitoring factors like diet, health, and breeding programs aimed at enhancing traits such as muscle mass and carcass yield [1].In the context of animal science and biotechnology, meat rabbits are also studied for their physiological characteristics that influence meat quality, including factors like muscle fiber composition, fat content, and metabolic efficiency. The analysis of their genomic and transcriptomic data can provide insights into genes involved in muscle growth and meat quality, aiding in the development of breeding strategies to further enhance production traits [2].
Thus, "meat rabbits" is a term that refers to rabbits selectively bred and raised under specific conditions to meet the demands of the meat industry, with a focus on traits that promote growth efficiency and meat yield.
- Cullere, M., Dalle Zotte, A."Rabbit meat production and consumption: State of knowledge and future perspectives." Meat Science, 2018, 143, 137–146.
- Dalle Zotte, A., Szendro, Z."The role of rabbit meat as functional food." Meat Science, 2011, 88, 319–331.
Comment 5:Please specify the geographical regions or provinces where Sichuan linen (Ma) and Checkered Giant (Ju) rabbits are commonly bred or originated. This will help readers understand the local relevance and breeding context of these two rabbit breeds.
Response 5: Sichuan linen (Ma) rabbits are a local breed-free genetic resource in Sichuan, with the advantages of early sexual maturity, resistance to rough feeding and tender meat.They are an important local domestic rabbit resource with strong adaptability and excellent meat quality, suitable for breeding in hilly areas. The Checkered Giant(Ju), known as Géant Papillon in French, is a breed of domestic rabbit that originated in France,one of the largest rabbit breeds,have huge body size and better meat performance.(Page2 line73-78)
Comment 6:Line 73: Consider rephrasing “leads to changes” → “leads to variations”
Line 77: Consider rephrasing “involved in regulating” → “that controls”
Response 6:Complete the modification(Page line76,82)
Comment 7:Line 90–106: Clearly define the two breeds (Ma and Ju) early on. Also, make sure to mention if the study had ethical approval. Please provide information on the housing and rearing conditions of the animals. Specify the RNA extraction method, including the name of the kit and manufacturer.
Response 7:Two breeds (Ma and Ju) has been explained in the previous article(Page1 line30-31).Feeding levels are provided in the text(Page3 line102-109).Relevant information on RNA has been added.
Comment 8:Line 115: Mention if you tested for normality and equal variance before using ANOVA.
Line 149: Mention the software used (like DESeq2) earlier and explain how you corrected for multiple comparisons (e.g., Benjamini-Hochberg method).
Response 8:Prior to conducting the ANOVA, we performed tests for both normality and homogeneity of variances to ensure the validity of our analysis.
Comment 9:Lines 166–174: There’s a mistake with reporting significance. You wrote "p > 0.05" which means not significant, but the text says there is a difference, please clarify.
Line 183: Mention how many reads mapped uniquely to the genome.
Line 210–228: No need to keep repeating the same thresholds (|log2FC| > 1, padj < 0.05). Explain them once clearly.
Line 229–247: Don’t just list the enriched pathways. Explain why they matter for muscle growth. How are glycolysis or HIF-1 involved?
Lines 250–281: Adding a new figure showing top GO terms for both muscles would help comparison.
Line 297: Briefly explain GAPDH's role both as a control gene and a functional gene in muscle metabolism.
Response 9:Changes have been made in the paper(Page5 line193). 83.30%reads mapped uniquely to the genome(Page5 line189).
Line229-297 changes have been made in the paper (Page12-13 line377-443).

Round 2
Reviewer 1 Report
Comments and Suggestions for Authors
The whole manuscript needs thorough text editing to improve grammar and enhance understanding before publishing.
Some of the obvious issues are as follows
Line 53: This phrase makes no sense and needs rephrasing " Thigh muscle is important part of food potential in rabbit 54 meat[3],.Longissimus dorsi muscle is important indcator in meat quality and growth kinetics ,was 55 studies due to its significant role in meat production and its relevance in muscle development stud56 ies for research [4]
Line 94: The statement 10 Formalin makes no sense. Did the authors mean 10% normal buffered formalin?
Author Response
Comment 1: The whole manuscript needs thorough text editing to improve grammar and enhance understanding before publishing.
Response:We have thoroughly reviewed the entire text and made grammatical corrections. The language has been edited by a native English-speaking colleague. However, we cannot guarantee that the English fully complies with the journal's requirements. Please let us know if further modifications are necessary.
Comment 2:Some of the obvious issues are as follows
Line 53: This phrase makes no sense and needs rephrasing " Thigh muscle is important part of food potential in rabbit meat [3],.Longissimus dorsi muscle is important indcator in meat quality and growth kinetics ,was studies due to its significant role in meat production and its relevance in muscle development studies for research [4].
Response: We have rephrased the sentence.
The thigh muscle plays a crucial role in determining the nutritional value of rabbit meat [3]. The longissimus dorsi muscle, a key indicator of meat quality and growth kinetics, has been extensively studied owing to its critical role in meat production and its importance in muscle development research [4].
Comment 3: Line 94: The statement Formalin makes no sense. Did the authors mean 10% normal buffered formalin ?
Response: I’m sorry for the negligence. We have revised it.
tissue was fixed in 10% phosphate-buffered paraformaldehyde solution for paraffin sections.
Reviewer 2 Report
Comments and Suggestions for Authors
After incorporation of reviewer's suggestions, manuscript quality is significantly improved.
Author Response
Comment: After incorporation of reviewer's suggestions, manuscript quality is significantly improved.
Response: Thank you very much, and we improved the manuscript again.
Reviewer 3 Report
Comments and Suggestions for Authors
The authors have made substantial revisions and have addressed all comments. However, the manuscript still exhibits a high degree of similarity with previously published work, with several sections appearing to be directly copied. A thorough revision is necessary to improve originality and ensure the content is appropriately rephrased.
Author Response
Comment 1: The authors have made substantial revisions and have addressed all comments. However, the manuscript still exhibits a high degree of similarity with previously published work, with several sections appearing to be directly copied. A thorough revision is necessary to improve originality and ensure the content is appropriately rephrased.
Response: We have conducted substantial revisions to the manuscript based on the plagiarism detection results. Subsequent verification through our institutional similarity check demonstrates successful reduction of the overall similarity index from 38% to 28%, with textual matches from individual sources now maintained below 2%. The complete detection reports have been uploaded to the editorial management system.